# Impact of a Carboxymethyl Cellulose Coating Incorporated with an Ethanolic Propolis Extract on the Quality Criteria of Chicken Breast Meat

**DOI:** 10.3390/antiox11061191

**Published:** 2022-06-17

**Authors:** Aly Farag El Sheikha, Ayman Younes Allam, Tahra ElObeid, Elham Abdelrahman Basiouny, Ahmad Abdelkaway Abdelaal, Ryszard Amarowicz, Emel Oz, Charalampos Proestos, Emad Karrar, Fatih Oz

**Affiliations:** 1College of Bioscience and Bioengineering, Jiangxi Agricultural University, 1101 Zhimin Road, Nanchang 330045, China; 2Faculty of Health Sciences, School of Nutrition Sciences, University of Ottawa, 25 University Private, Ottawa, ON K1N 6N5, Canada; 3Department of Food Science and Technology, Faculty of Agriculture, Minufiya University, Shibin El Kom 32511, Egypt; ayman_alaam@yahoo.com; 4Human Nutrition Department, College of Health Sciences, QU Health, Qatar University, Doha P.O. Box 2713, Qatar; tahra.e@qu.edu.qa; 5Economic Entomology & Agriculture Zoology Department, Faculty of Agriculture, Menofiya University, Shibin El Kom 32511, Egypt; eabdo1169@gmail.com (E.A.B.); aakaway@yahoo.com (A.A.A.); 6Institute of Animal Reproduction and Food Research, Polish Academy of Sciences, 10-748 Olsztyn, Poland; 7Department of Food Engineering, Faculty of Agriculture, Ataturk University, Erzurum 25240, Turkey; emel.oz@atauni.edu.tr; 8Laboratory of Food Chemistry, Department of Chemistry, School of Sciences, National and Kapodistrian University of Athens, 15772 Athens, Greece; harpro@chem.uoa.gr; 9National Engineering Research Center for Functional Food, Collaborative Innovation Center of Food Safety and Quality Control in Jiangsu Province, School of Food Science and Technology, Jiangnan University, Wuxi 214122, China; emadkarrar26@yahoo.com

**Keywords:** bioactive coating, carboxymethyl cellulose, propolis, physicochemical and microbiological quality parameters, sensory evaluation

## Abstract

Recently, the demand for composite edible coatings has increased significantly as a new trend to confront the serious processing and storage problems that always arise regarding chicken meat. We aim to develop a carboxymethyl cellulose (CMC) coating containing various concentrations (0, 1, 2, 3, and 4%) of an ethanolic propolis extract (EPE) to maintain the quality and extend the shelf life of chicken breast meat stored at 2 °C for 16 days. The influence of the CMC and EPE coating on the physicochemical and microbiological quality parameters of chicken breast meat, e.g., pH, color, metmyoglobin (MetMb), lipid oxidation (thiobarbituric acid reactive substance, TBARS), and microbiological and sensory analyses, was studied. Significantly lower weight loss and pH (*p* ≤ 0.05) were noted in the coated samples compared with the uncoated samples (control) over the storage period. MetMb content was significantly reduced (*p* ≤ 0.05) in the coated samples compared to the control. Additionally, the addition of EPE to CMC was more effective in inhibiting microbial growth, preventing lipid oxidation, and keeping the overall acceptability of coated chicken breast meat compared to the control. This work presents CMC and EPE as alternative preservatives to produce active packaging coatings.

## 1. Introduction

Chicken is one of the most popular types of meat. Chicken meat is a great source of many key nutrients, including protein, vitamins, and minerals [1,2,3]. However, the chicken comes in a variety of cuts, including breasts, thighs, wings, and drumsticks. Each cut differs regarding its distinct content of nutrients [4]; for example, chicken breast is characterized by its high protein content (especially lean protein), in addition to its low content of sodium and being devoid of carbohydrates (sugar-free), which reflects positively its health benefits [5]. As with other types of meat, chicken meat is easily and rapidly damaged (highly perishable), and the high consumption of poultry meat products leads to concerns pertaining to safety, quality, and linked sensory properties [6]. Despite the technological boom in recent years in meat preservation and processing, crises still surround poultry meat, whether fresh or during processing or storage, making it a permanent and growing source of danger. As chicken meat is a natural reservoir for *Salmonella*, recently, a multistate *Salmonella* outbreak was traced to raw, frozen, breaded, and stuffed chicken products in the USA and Qatar. Hence, the USDA issued a public alert, but no recalls have been initiated since the posting of the outbreak announcement [7,8,9]. Unsaturated lipids, environment light, fine grinding, the incorporation of air, metal contact, and high temperatures during processing contribute to lipid oxidation. After microbial deterioration, lipid oxidation is the primary process that results in the loss of meat quality [10]. Although freezing, chilling, and cold storage are effective methods for preserving meats, e.g., chicken meat, they cannot completely prevent chemical and oxidative reactions [11]. Promising natural antioxidant compound sources are the meat industry’s new strategy to extend the shelf life of meat and meat products [12].

Propolis extract is a natural compound found in every honey beehive and is a stable, natural preservative, antimicrobial, antioxidant, and odorless substance. It also positively impacts human health [13,14]. Thus, according to previous research, this product is an excellent source of antioxidants associated with cinnamic, flavonoids, caffeic acids, and benzoic acids, either singularly or in synergistic combinations [15,16,17]. Additionally, propolis has antimicrobial activities against various microbes [18,19,20]. Natural preservatives’ benefits have recently been enhanced by incorporating them into various edible coatings and films on food products [21,22]. Recently, interest has risen in using bio-based substances in producing bioactive coatings, i.e., edible coatings that are applied in chicken breast packaging. The use of agricultural by-products and wastes (which are significant sources of polysaccharides, e.g., agar, chitosan, cellulose, and starch) constituted a boom and revolution not only in the field of extracting bio-based materials, but also to find an eco-friendly solution for the disposal of these wastes [23,24,25]. Carboxymethyl cellulose (CMC) is a natural polysaccharide and a hydrophilic polymer used in food coatings and films [26,27]. Coatings with CMC could be used to increase the shelf life and maintain the quality of food by forming an oxygen barrier, reducing water loss, preventing microbiological growth, and decreasing weight loss. Additionally, they protect the organoleptic characteristics of food by inhibiting and retarding lipid and protein oxidation.

Due to the benefits of propolis as a natural preservative (antioxidant and antimicrobial), it could be incorporated with an edible coating to improve the quality and extend the shelf life of meat. There is no published data on the application of CMC/propolis composite films to improve the quality and shelf life of meat and meat products. Therefore, to our knowledge, this is the first paper to study the effects of using a composite edible coating from CMC containing various levels (0, 1, 2, 3, and 4%) of an ethanolic propolis extract (EPE) on the physicochemical, chemical, microbiological, and sensory quality parameters of chicken breast meat during refrigerated storage.

## 2. Materials and Methods

### 2.1. Materials

Propolis was taken from an apiary around Shibin El-Kom City, Menoufia, Egypt, and kept frozen (−18 °C) until needed. Fresh chicken breast fillets were purchased from a local market (Shibin El-Kom, Egypt) from leghorn chicken female birds (*Gallus gallus domesticus*) slaughtered at 9 weeks of age when they weigh 5.5 pounds. Chicken breast meat samples were placed on ice in a polystyrene box and transported to the laboratory within one hour of the slaughter. All samples were cut aseptically and packaged into different sizes in the range of 25–50 g for use in treatment preparation.

### 2.2. Chemicals and Reagents

CMC, glycerol, ethanol, methanol, and molten plate count agar were obtained from Sigma-Aldrich (St. Louis, MO, USA). Folin–Ciocalteu reagent (FCR), trichloroacetic acid (TCA), gallic acid (GA), trolox,1,1-diphenyl-2-picrylhydrazyl (DPPH), thiobarbituric acid (TBA), and quercetin, butylated hydroxytoluene (BHT) were procured from Sigma-Aldrich Chemie GmbH (Eschenstr, Taufkirchen, Germany).

### 2.3. Preparation of Ethanolic Propolis Extract (EPE)

The EPE solution was made by mixing 100 g of powdered crude propolis (100 g of frozen crude propolis kept at −18 °C was grinded in a mortar until a powder was obtained [28]) with 70% ethanol (*w*/*v*) to obtain a final volume of 1000 mL. The macerate was then incubated at 30 °C for seven days, stirring at least once per hour. The solutions were filtered through Whatman No. 4 filter paper. The resultant residue was evaporated in a rotary evaporator (under vacuum) at 45 °C until it reached an 80–85% volume reduction. The total phenolic content (TPC) and radical scavenging ability (DPPH) of the resulting EPE were estimated using the procedures of Wieczyńska et al. [29].

### 2.4. Preparation of the Coating Films

The films were prepared as described by Dashipour et al. [30] with some modifications. The CMC solution was prepared by dissolving 1 g CMC in 100 mL distilled water (1% *w*/*v*) under constant magnetic stirring at 55–60 °C for 50 min until its complete dissolution was achieved. Afterwards, 0.5 mL glycerol (0.5% *w*/*v* based on the CMC) was added, and the mixture was heated at 85 °C and stirred continuously for 5 min. Then, the mixture was cooled to room temperature. Different percentages (0, 1, 2, 3, and 4%) of EPE were added to CMC to prepare the composite film coatings (CMC-P0%, CMC-P1%, CMC-P2%, CMC-P3%, and CMC-P4%).

### 2.5. Chicken Breast Meat Coating Process

Each of the six chicken breast’s deboned parts was processed to remove the exterior fat, and then cut into slices. The first part was soaked in distilled water to prepare the control samples (uncoated). The other five parts were soaked in the CMC/EPE composite coating solution for two minutes to prepare the five coating treatments (CMC-P0%, CMC-P1%, CMC-P2%, CMC-P3%, and CMC-P4%). The residual coating solution was removed by allowing the samples to dry for another 3 minutes under fresh airflow. The uncoated and coated samples were packed in a styrofoam box (trays) with a polyvinyl chloride elastic strap and kept in the refrigerator at 2 °C for 0, 4, 8, 12, and 16 days before further analysis.

### 2.6. Physicochemical Analyses

#### 2.6.1. Analysis of pH Value

The pH values of chicken breast meat samples were determined using a digital pH meter (Model 3510, Jenway Technology, Milano, Italy), according to the method described by Feldsine et al. [31].

#### 2.6.2. Weight Loss (WL%)

The change in meat weight during storage (before/after) was used to estimate the weight loss (WL%) of meat samples as described by Campañone et al. [32] using the following equation: (1)WL (%)=(W0− Wt)×100/W0
where W0 is the meat sample’s weight (g) before storage, and Wt is the meat sample’s weight (g) after 4, 8, 12, and 16 days.

#### 2.6.3. Color Assessment

The instrumental color (L*, a*, and b*) analysis of raw chicken meat samples was conducted using a scale color spectrophotometer (machine colors Tristimulus) with a CIELab colorimeter (Hunter, LabScan^®^ XE, Reston, VA, USA), considering ‘Daylight 65’ as the standard illuminant with a 2° standard observer and lighting area of 8 mm according to the method of Kandil et al. [33] and Garavito et al. [34]. The color difference (ΔE*) between the control and coated samples containing EPE was calculated using the following equation:(2)$${\Delta E}^{*}=\sqrt{{\Delta L*}^{2\;}+{\;\Delta a*}^{2\;}+{\Delta b*}^{2\;}}$$
where ΔL* is the brightness difference, Δa* is the redness difference, and Δb* is the yellowness difference.

#### 2.6.4. Metmyoglobin (MetMb) Content Determination

The MetMb content in chicken was estimated by the method described by Lindahl et al. [35], taking into consideration the modifications that were made by Tang [36].

#### 2.6.5. Assessment of Lipid Oxidation (Thiobarbituric Acid Reactive Substances, TBARS)

The TBARS values of chicken meat samples were determined spectrophotometrically according to Peiretti et al. [37]. The absorbance was measured at 530 nm using a spectrophotometer (Model UV-VIS-2802PC, New Jersey, USA). The TBARS values were stated as milligrams of malonaldehyde per kg of chicken meat samples.

### 2.7. Microbiological Analysis

The total aerobic plate count (TAPC) was determined according to the procedure of Horváth [38] using molten plate count agar (MPCA). Plates were incubated at 30 °C for 72 h. The obtained data were transformed into a logarithm of colony-forming units per gram of chicken meat (log CFU/g). Triplicate measurements were performed.

### 2.8. Sensory Evaluation

A sensory evaluation of cooked chicken meat samples (microwave oven for six min) was conducted by 25 trained panelists of staff members (aged 21–40 years) of the Department of Food Science and Technology, Faculty of Agriculture, Menofiya University, according to the method described by Kandil et al. [33]. Panelists were selected based on their interests and availability. The panelists were asked to rate the cooked samples’ color and odor using a scale point ranging from 0 to 10, where 10 = excellent; 9 = very good; 8 = good; 7 = acceptable; 6 = poor. The product was defined as unacceptable after the onset of a bad odor or unpleasant taste. The fresh chicken breast meat was used as a reference.

### 2.9. Statistical Analysis

The study was replicated three times, and three measurements were conducted for each replicate. The data were analyzed using the SPSS program. The mean values of different parameters were used to compare pH values, DPPH, total phenolic content, color values, and sensory characteristics. Analysis of variance (ANOVA) was used to analyze microbial data. The means were separated with the least significant difference procedure when a significant effect was detected. A two-way analysis of variance was used for multiple variable comparisons. Significance between groups was conducted using Duncan’s analysis (*p* ≤ 0.05).

## 3. Results and Discussion

### 3.1. Test Probabilities for Physicochemical, Microbiological, and Sensory Criteria of Chicken Meat Samples Generally and Depending on Storage Time and the Coating Treatment—Multi-Aspect Variance Analysis Including Interactions

The storage time and the coating treatment of the chicken breast meat samples can have a significant impact on their physicochemical and microbiological characteristics. The data presented in Table 1 show a significant effect (*p* < 0.001) of storage period on all measured parameters except the overall acceptance, for which the significant effect is *p* < 0.01. Additionally, the storage period had a smaller effect on color (*p* < 0.05). Furthermore, Table 1 illustrates a significant effect (*p* < 0.001) of coating treatment on all measured parameters, except TBARS and color characteristic (a* value), in which the significant effect is *p* < 0.01. Moreover, the coating treatment had a smaller effect on the total color difference (∆E) (*p* < 0.05). There was no effect of coating treatment on color characteristic (b* value), metmyoglobin (MetMb) content, and total aerobic plate count (TAPC). The interactions between the storage time and the coating treatment were also indicated for all the tested parameters. The storage time and the coating treatment had a significant effect (*p* < 0.001) on all measured parameters.

### 3.2. Physicochemical Analyses of the Chicken Breast Meat Samples

#### 3.2.1. pH Values

Chicken breast meat’s quality, as well as its gastronomic and manufacturing compatibility were determined by its pH value. Under normal conditions, fresh chicken meat has a pH range of about 5.3–6.5 [1]. In this investigation, chicken meat with pH 5.51 and pH 5.87 values (typical of normal flesh), but does not have unfavorable sensory characteristics, such as pale, soft, and exudative (PSE) or dark, firm, and dry (DFD), was employed. The pH values of the preserved chicken breast meat were significantly affected by storage time (ST) and treatment (T) (*p* ≤ 0.05) (see Figure 1). Starting on day 4 and continuing until the end of storage, the pH values of coated chicken breast meat samples with CMC and EPE were lower (*p* ≤ 0.05) than those of uncoated samples (control). The limited spectrum of proteolytic alterations in muscle proteins, resulting in the progressive alkalization of the refrigerated chicken, could explain the pH differences observed between the coated and uncoated samples (controls). Furthermore, on days 8, 12, and 16, the samples of chicken breast meat coated with CMC plus 4% propolis (CMC-P4%) extract had lower (*p* ≤ 0.05) pH values than the samples of chicken breast meat coated only with CMC. Propolis’ antimicrobial and antioxidant properties may be responsible for the observed pH changes in stored chicken breast meat, preventing proteolysis and microbiological development changes. Furthermore, the pH values of the coated and uncoated chicken breast meat samples increased as the storage period increased. At the end of the storage period, the increase in pH values of the uncoated samples (controls) was more pronounced. This can happen due to the accumulation of ammonia and amino acid degradation products, which causes the pH to rise [39]. An increase in the pH values of stored meat may be linked to the production of peptides, amino acids, and ammonia due to increased protease activity (cathepsins B, calpains, and L peptidases) or microbial development [40,41].

#### 3.2.2. Weight Loss (WL%)

At all interval storage periods, chicken meat coated with CMC and EPE lost less weight (*p* ≤ 0.05) than the uncoated samples (controls) (Figure 2). There were no significant (*p* ˃ 0.05) differences in weight loss among the coated chicken breast meat samples with different levels of EPE after 4 days of refrigerated storage. However, after 8, 12, and 16 days of storage, the samples of chicken breast meat coated with 4% EPE showed a minor weight loss (*p* ≤ 0.05) at all EPE levels. Furthermore, the weight loss for the coated and uncoated chicken breast meat samples increased as the storage period increased. The formation of CMC films, which tend to cluster water inside their matrix material and polymer, could explain the reduction in weight loss for the samples of chicken breast meat coated with CMC and EPE. CMC films may also have high oxygen barrier properties. According to Bodini et al. [42], Ulloa et al. [43], and Yong and Liu [44], adding propolis extract to coating films can reduce water vapor penetration. The amount of water evaporation and condensation as well as the leaking of chicken breast meat juice due to microbial deterioration and hydrolysis reactions in the presence of oxygen influenced the amount of weight loss of chicken during refrigeration storage [45].

#### 3.2.3. Color Assessment

Figure 3A–D shows the effects of coating chicken breast meat with CMC containing various levels of EPE on color characteristics. Both the coating with CMC containing EPE and storage time had a significant (*p* ≤ 0.05) impact on all color characteristics (L*, a*, b*, and ∆E*) of the stored chicken breast meat samples. After 4 days of refrigerated storage, the samples of chicken breast meat coated with CMC and EPE had similar (*p* ˃ 0.05) lightness to the uncoated samples (control), except for the samples of chicken breast meat coated with 4% EPE, which had more (*p* ≤ 0.05) lightness than the control (Figure 3A). The uncoated chicken breast meat samples showed a darker (*p* ≤ 0.05) color than the coated chicken meat samples with CMC and EPE at 1, 2, 3, and 4% levels as the storage period increased between 12 and 16 days. The oxidation of oxymyoglobin to metmyoglobin may be responsible for the dark color of the uncoated samples (Figure 3A).

After 4, 8, 12, and 16 days of refrigerated storage, the samples of chicken breast meat coated with CMC containing 4% EPE had higher (*p* ≤ 0.05) redness than the uncoated samples (control) (Figure 3B). In addition, the redness of the coated and uncoated samples decreased as the storage period progressed. The uncoated sample (control) had the highest reduction in redness at the end of the refrigerated storage. In contrast, the samples of chicken breast meat coated with CMC containing 4% EPE had the lowest reduction in redness. According to Ruelas-Chacon et al. [46], chitosan films treated with propolis extract significantly reduced redness changes in stored Nemipterus japonicus pieces, which coincide with our results. At all levels of EPE and CMC, the coated chicken breast meat samples had a lower (*p* ≤ 0.05) yellowness than the uncoated samples at all storage intervals (Figure 3C). The yellowness of the coated and uncoated chicken meat increased as the storage time progressed.

The color difference changes (∆E*) of the coated chicken increased (*p* ≤ 0.05) as the storage period increased (Figure 3D). The color difference change of the samples of chicken breast meat coated with CMC containing 4% EPE was more pronounced (*p* ≤ 0.05) than in the other coated samples, especially after 8, 12, and 16 days of refrigerated storage.

#### 3.2.4. Metmyoglobin (MetMb) Content

The percentage concentration of MetMb (Figure 4) increased steadily (*p* ≤ 0.05) in all groups studied as the storage period increased. Nonetheless, hydrogen bonding in CMC films is thought to reduce oxymyoglobin (red) to MetMb (brown) oxidation and limit oxygen access to meat [47]. This is consistent with our findings, which show that throughout all storage periods, the samples of chicken breast meat coated with CMC containing EPE ranging from 2% to 4% had a lower (*p* ≤ 0.05) MetMb content than the uncoated chicken breast meat samples (control).

According to the current study’s findings, when EPE is used in conjunction with CMC coatings, there is a significant reduction (*p* ≤ 0.05) in MetMb synthesis and formation. Ruan et al. [48] reported similar findings, claiming that CMC coatings infused with epigallocatechin gallate and sodium alginate could prevent MetMb levels from developing in fresh pork stored at 4 ± 1 °C for 7 days. This is most likely because EPE affected the effectiveness of the oxidation process in stored meat and inhibited the aerobic plate count (TAPC) as explained later.

#### 3.2.5. Lipid Oxidation

As the storage period progressed, TBARS levels in coated and uncoated chicken meat increased, with the control chicken having the highest (*p* ≤ 0.05) TBARS value at each interval storage period (Figure 5). The samples of chicken breast meat coated with CMC containing EPE at 2, 3, and 4% levels had lower (*p* ≤ 0.05) TBARS values compared to the uncoated chicken breast meat samples (controls). The low permeability of the CMC coating may explain why coated chicken breast meat does not oxidize lipids during refrigerated storage. Despite this, the oxygen permeability and penetration of the CMC coating film increase during storage due to the relationship between CMC molecules and water in high-moisture foods, such as meat. In these cases, adding the right amount of phenolic chemicals and free radical scavengers to CMC solutions could be crucial in preventing oxidation in the coated samples.

At the end of the storage period, the samples of chicken breast meat coated with CMC and EPE at a 4% level reduced the rate of oxidative rancidity in the preserved chicken breast meat samples, resulting in the lowest (*p* ≤ 0.05) TBARS value, whereas the control and coated chicken with CMC only had the highest (*p* ≤ 0.05) TBARS values. Furthermore, the chicken coated with CMC and EPE at 2, 3, and 4% levels had lower (*p* ≤ 0.05) TBARS values than the samples of chicken breast meat coated with CMC only at each interval storage period. This could be due to the antioxidant properties of propolis, which are caused by the presence of polyphenolic substances that were estimated in our study by 921.82 ± 11.75 mg (gallic acid equivalent/100 g) and 92.25 ± 2.34% radical scavenging activity, respectively. Similar observations were obtained by Ebadi et al. [49], who found that adding propolis extracts to chitosan films at a concentration of 2–3% reduced TBARS readings in preserved pork chops. Additionally, this conforms with what was reported by Santos et al. [50] that the antimicrobial and antioxidant properties attributed to propolis are valuable for the food industry due to their effect on delaying lipid oxidation and improving food shelf life.

### 3.3. Total Aerobic Plate Count (TAPC)

The current study found that storage time and coatings with CMC and EPE impacted the TAPC of chicken meat (Figure 6). TAPC levels in chicken meat should not exceed 7 log CFU/g, according to the recommended upper limit [51,52].

As the storage period progressed, the TAPC of coated and uncoated chicken breast meat samples increased significantly (*p* ≤ 0.05), but the numbers remained within the recommended range. Only the control sample count approached the spoil limit at the end of storage. At each interval storage period, the TAPC of coated chicken breast meat samples with CMC containing EPE was significantly lower (*p* ≤ 0.05) than that of the control. This was primarily because CMC coatings have low gas permeability, which could effectively inhibit aerobic organism growth. Furthermore, the samples of chicken breast meat coated with CMC plus 1, 2, 3, and 4% propolis (CMC-P1, P2, P3, and P4%) extract had lower (*p* ≤ 0.05) TAPC values than the samples of chicken breast meat coated only with CMC on days 4, 8, 12, and 16. The antimicrobial properties of propolis may be responsible for lowering TAPC and preventing microbiological development in refrigerated storage. The chicken breast meat coated with CMC and 4% EPE had the lowest (*p* ≤ 0.05) TAPC of all the treatments at the end of the storage period, while the uncoated chicken breast meat samples (controls) had the highest (*p* ≤ 0.05) TAPC. Rezaei and Shahbazi [53] also found that adding propolis to a CMC matrix inhibited microbiological development in refrigerated fish fillets. These effects are primarily due to propolis extract’s antibacterial properties caused by polyphenolic content, inhibiting RNA and DNA synthesis by altering microorganism cell membranes [54].

Furthermore, our findings are consistent with those of Shavisi et al. [55], who discovered that a polylactic acid (PLA) film containing propolis ethanolic extract extended the shelf life of minced beef during storage in refrigerated conditions for at least 11 days without any unfavorable organoleptic properties.

### 3.4. Sensory Evaluation

Storage time and treatment had a significant impact (*p* ≤ 0.05) on the sensory scores of the samples of chicken breast meat coated with CMC and EPE at various levels (Figure 7A–C). As the storage time increased, the color, odor, and overall acceptability scores significantly decreased. At 4, 8, and 12 days of refrigerated storage, the samples of chicken breast meat coated with CMC containing 3 and 4% EPE had lower (*p* ≤ 0.05) color rating scores than the chicken breast meat coated with CMC containing 0, 1, and 2% EPE, as well as the control (Figure 7A). When compared to the samples of chicken breast meat coated with CMC containing 0, 1, 2, and 3% EPE on day 8 of refrigerated storage, the samples of chicken breast meat coated with CMC and 4% EPE had a negative impact on the odor score (*p* ≤ 0.05) (Figure 7B). Furthermore, the samples of chicken breast meat coated with CMC and 4% EPE had a more offensive odor (*p* ≤ 0.05) than the control chicken breast meat samples. On day 12 of storage, the samples of chicken breast meat coated with CMC and EPE at 0%, 1%, and 2% levels had higher rating odor scores than the control chicken breast meat samples, but the scores were within panelists’ acceptable range. On day 8 of storage, the overall acceptability of the samples of chicken breast meat coated with CMC and 4% EPE was lower than (*p* ≤ 0.05) that coated with CMC containing 0, 1, 2, and 3% EPE. Furthermore, on day 12 of storage, the overall acceptability of the samples of chicken breast meat coated with CMC and EPE at 0, 1, and 2% levels were higher than (*p* ≤ 0.05) of the control (Figure 7C). The panelists found the overall acceptability scores to be within acceptable limits. Our results are consistent with what was seen by Shavisi et al. [55], who mentioned the positive impact of adding a propolis ethanolic extract to a PLA film on the shelf life of minced beef without any unfavorable organoleptic for 11 days of cold storage. Pastor et al. [56] and Palou et al. [57] also stated that the natural flavor of propolis extract should be considered when choosing the propolis concentration in coating systems.

## 4. Conclusions

From the above results, it can be concluded that coating chicken breast meat with carboxymethyl cellulose containing 3% of ethanolic propolis extract (because the coating had 4% propolis extract may cause off-odor) and storing it in refrigerated conditions could preserve the physicochemical, microbiological, and sensory quality criteria and extend the shelf life of chicken breast meat. This work presented a composite edible coating from CMC and EPE as a potent natural alternative to synthetic preservatives to produce active packaging coatings applicable for several food types.

## Figures and Tables

**Figure 1 antioxidants-11-01191-f001:**
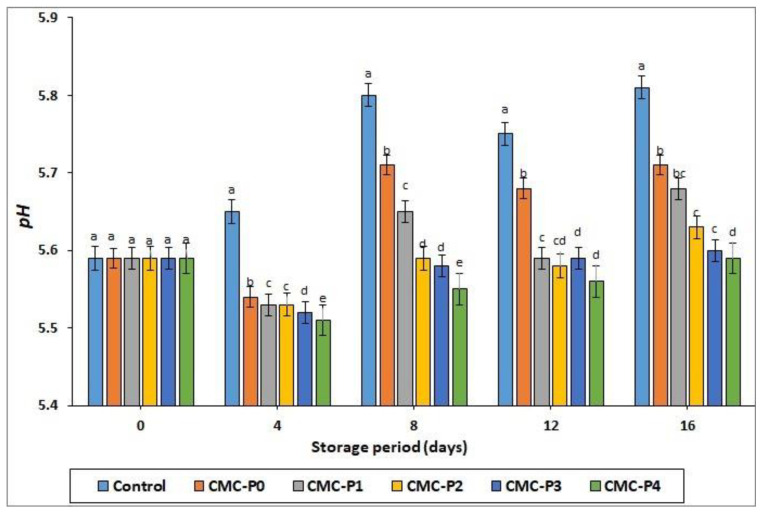
The impact of applied coatings on the pH values of chicken breast meat samples during storage at 2 °C for 16 days. **Control**: Uncoated chicken breast meat samples (soaked samples in sterile distilled water). **CMC-P0**: CMC-coated chicken breast meat samples with 0% EPE. **CMC-P1**: CMC-coated chicken breast meat samples with 1% EPE. **CMC-P2**: CMC-coated chicken breast meat samples with 2% EPE. **CMC-P3**: CMC-coated chicken breast meat samples with 3% EPE. **CMC-P4**: CMC-coated chicken breast meat samples with 4% EPE. ^a–e^: Within a column, different superscripts indicate significant differences (*p* ≤ 0.05).

**Figure 2 antioxidants-11-01191-f002:**
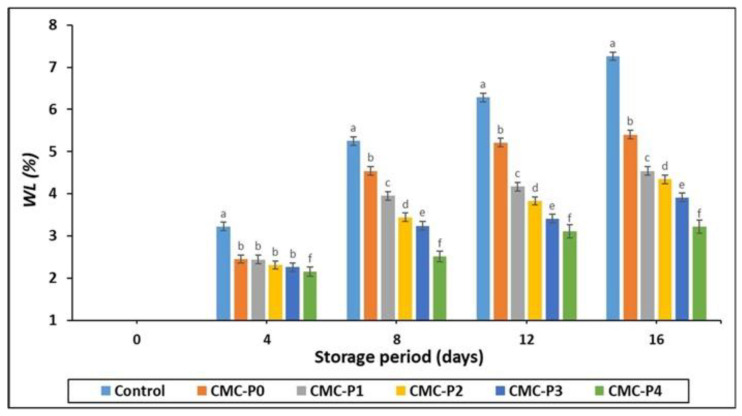
The impact of applied coatings on the WL (%) of chicken breast meat samples during storage at 2 °C for 16 days. **Control**: Uncoated chicken breast meat samples (soaked samples in sterile distilled water). **CMC-P0**: CMC-coated chicken breast meat samples with 0% EPE. **CMC-P1**: CMC-coated chicken breast meat samples with 1% EPE. **CMC-P2**: CMC-coated chicken breast meat samples with 2% EPE. **CMC-P3**: CMC-coated chicken breast meat samples with 3% EPE. **CMC-P4**: CMC-coated chicken breast meat samples with 4% EPE. ^a–f^: Within a column, different superscripts indicate significant differences (*p* ≤ 0.05).

**Figure 3 antioxidants-11-01191-f003:**
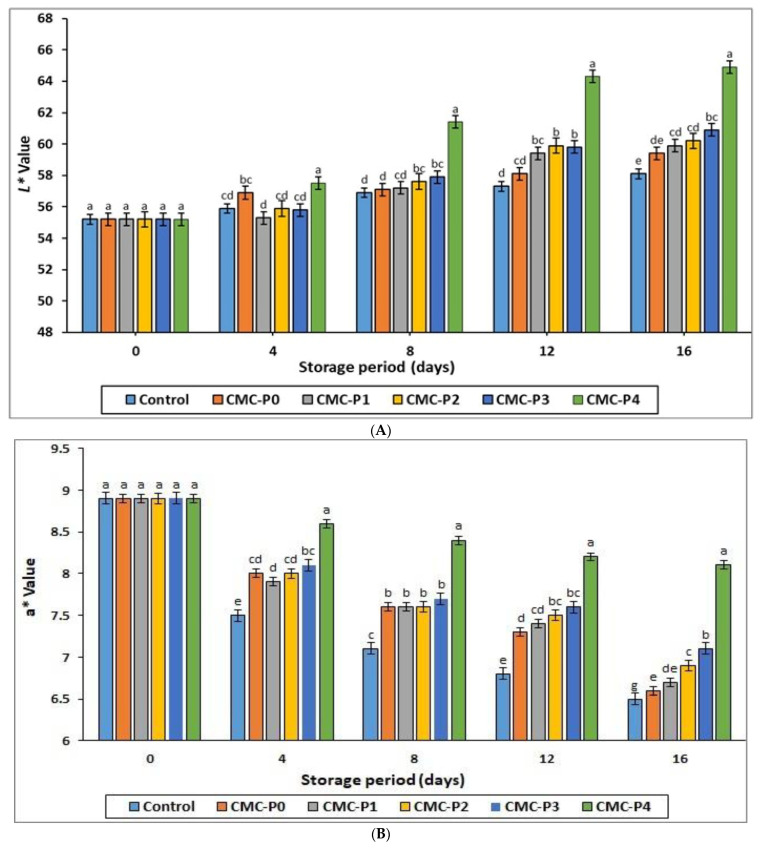
The impact of the applied coatings on all color characteristics: (**A**) L* value, (**B**) a* value, (**C**) b* value, and (**D**) ∆E (total color difference) of the chicken breast meat samples during storage at 2 °C for 16 days. **Control**: Uncoated chicken breast meat samples (soaked samples in sterile distilled water). **CMC-P0**: CMC-coated chicken breast meat samples with 0% EPE. **CMC-P1**: CMC-coated chicken breast meat samples with 1% EPE. **CMC-P2**: CMC-coated chicken breast meat samples with 2% EPE. **CMC-P3**: CMC-coated chicken breast meat samples with 3% EPE. **CMC-P4**: CMC-coated chicken breast meat samples with 4% EPE. ^a–g^: Within a column, different superscripts indicate significant differences (*p* ≤ 0.05).

**Figure 4 antioxidants-11-01191-f004:**
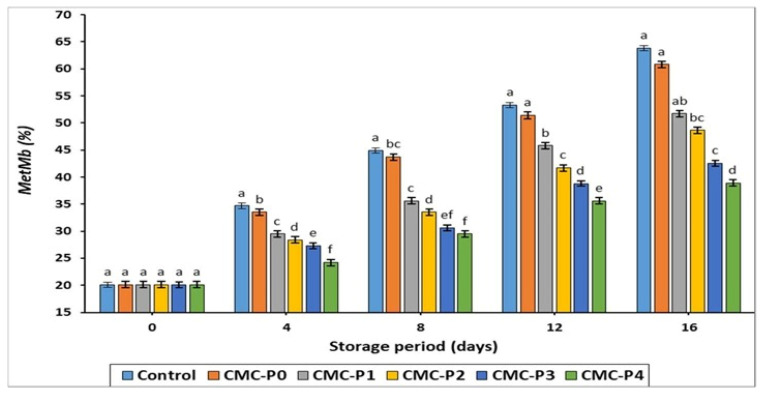
The impact of applied coatings on the MetMb (%) of chicken breast meat samples during storage at 2 °C for 16 days. **Control**: Uncoated chicken breast meat samples (soaked samples in sterile distilled water). **CMC-P0**: CMC-coated chicken breast meat samples with 0% EPE. **CMC-P1**: CMC-coated chicken breast meat samples with 1% EPE. **CMC-P2**: CMC-coated chicken breast meat samples with 2% EPE. **CMC-P3**: CMC-coated chicken breast meat samples with 3% EPE. **CMC-P4**: CMC-coated chicken breast meat samples with 4% EPE. ^a–f^: Within a column, different superscripts indicate significant differences (*p* ≤ 0.05).

**Figure 5 antioxidants-11-01191-f005:**
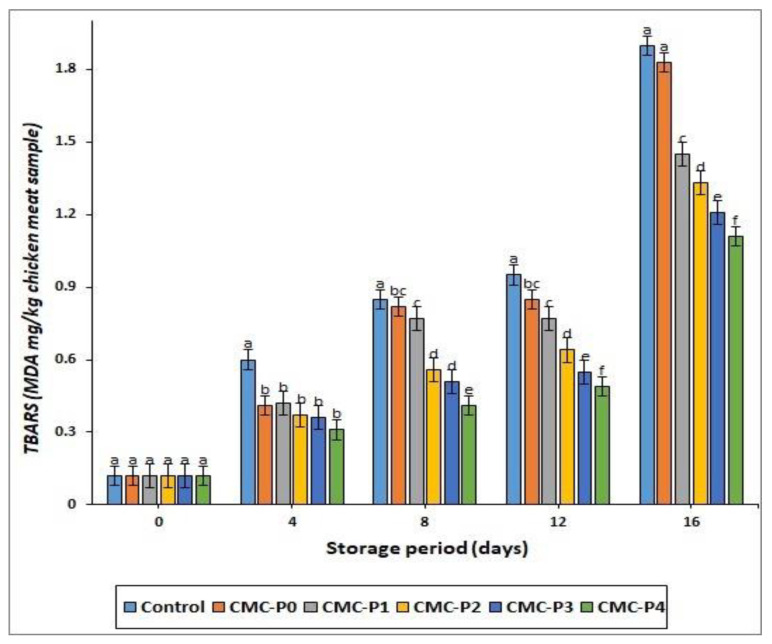
The impact of applied coatings on the TBARS values (MDA mg/kg chicken breast meat sample) of chicken breast meat samples during storage at 2 °C for 16 days. **Control**: Uncoated chicken breast meat samples (soaked samples in sterile distilled water). **CMC-P0**: CMC-coated chicken breast meat samples with 0% EPE. **CMC-P1**: CMC-coated chicken breast meat samples with 1% EPE. **CMC-P2**: CMC-coated chicken breast meat samples with 2% EPE. **CMC-P3**: CMC-coated chicken breast meat samples with 3% EPE. **CMC-P4**: CMC-coated chicken breast meat samples with 4% EPE. ^a–f^: Within a column, different superscripts indicate significant differences (*p* ≤ 0.05).

**Figure 6 antioxidants-11-01191-f006:**
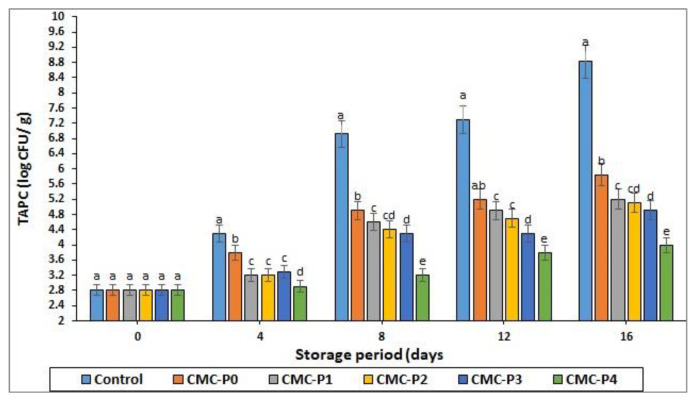
The impact of applied coatings on the TAPC of chicken breast meat samples during storage at 2 °C for 16 days. **Control**: Uncoated chicken breast meat samples (soaked samples in sterile distilled water). **CMC-P0**: CMC-coated chicken breast meat samples with 0% EPE. **CMC-P1**: CMC-coated chicken breast meat samples with 1% EPE. **CMC-P2**: CMC-coated chicken breast meat samples with 2% EPE. **CMC-P3**: CMC-coated chicken breast meat samples with 3% EPE. **CMC-P4**: CMC-coated chicken breast meat samples with 4% EPE. ^a–e^: Within a column, different superscripts indicate significant differences (*p* ≤ 0.05).

**Figure 7 antioxidants-11-01191-f007:**
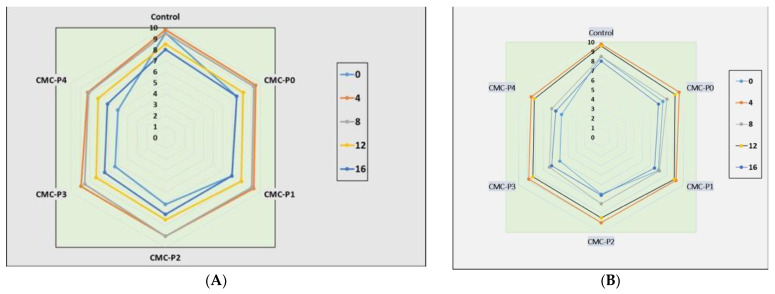
The impact of applied coatings on sensory evaluation: (**A**) color, (**B**) odor, and (**C**) overall acceptability of chicken breast meat samples during storage at 2 °C for 16 days. **Control**: Uncoated chicken breast meat samples (soaked samples in sterile distilled water). **CMC-P0**: CMC-coated chicken breast meat samples with 0% EPE. **CMC-P1**: CMC-coated chicken breast meat samples with 1% EPE. **CMC-P2**: CMC-coated chicken breast meat samples with 2% EPE. **CMC-P3**: CMC-coated chicken breast meat samples with 3% EPE. **CMC-P4**: CMC-coated chicken breast meat samples with 4% EPE.

**Table 1 antioxidants-11-01191-t001:** Test probabilities for physicochemical, microbiological, and sensory criteria of the chicken meat samples generally and depending on storage time and the coating treatment—multi-aspect variance analysis including interactions.

Criteria	Effect		Interaction T × ST
	Treatment (T)	Storage Time (ST)	
pH	XXX ^1^	XXX	XXX
Weight loss (%)	XXX	XXX	XXX
L*	XXX	XXX	XXX
a*	XX	XXX	XXX
b*	NS ^2^	XXX	XXX
∆E ^3^	X	XXX	XXX
TBARS	XX	XXX	XXX
MetMb	NS	XXX	XXX
TAPC	NS	XXX	XXX
Color	XXX	X	XXX
Odor	XXX	XXX	XXX
Overall acceptance	XXX	XX	XXX

^1^ X: *p* < 0.05; XX: *p* < 0.01; XXX: *p* < 0.001. ^2^ NS: not significant. ^3^ ∆E: Total color difference.

## Data Availability

Data is contained within the article.

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
