# Peer review of "Impact of a Carboxymethyl Cellulose Coating Incorporated with an Ethanolic Propolis Extract on the Quality Criteria of Chicken Breast Meat"

_antioxidants, 2022, doi:10.3390/antiox11061191_

Round 1

Reviewer 1 Report

The work presented was well designed, the text is well structured and easy to read. The results obtained are coherent and expected for works of this nature, having supported with previously published works.

Author Response

Responses to Reviewers’ Comments

First, many thanks to Editors and Reviewers for Recommending “Minor Revisions” to our manuscript for publication.

We provide upon resubmission:

Our revised manuscript was edited according to the rules of the JournalAntioxidants”. As requested, we activated the "Tracked Changes" option to highlight any changes in our paper.

The text was highlighted with yellow color to respond to some reviewers’ comments. Also, the text was highlighted in a turquoise color that mentioned the new additions.

As follows a clear list of our responses to Reviewers’ Comments, point by point.

Responses to Reviewer (1) Comments

  • The work presented was well designed, the text is well structured and easy to read. The results obtained are coherent and expected for works of this nature, having been supported by previously published works.

Responses

Firstly, many thanks to Reviewer (1) for describing our manuscript as “The work presented was well designed, the text is well structured and easy to read. The results obtained are coherent and expected for works of this nature”.

Also, we very much appreciated the recommendation of Reviewer (1) that “The results of our manuscript having been supported with previously published works”.

Reviewer 2 Report

The paper is well writen, documented and the experimental part is meaningfull. Some aspects require rectification: table 1 is unreadable and it is usefull to present soome images with treated meat for evidentiation of coating effect

Author Response

Responses to Reviewers’ Comments

First, many thanks to Editors and Reviewers for Recommending “Minor Revisions” to our manuscript for publication.

We provide upon resubmission:

Our revised manuscript was edited according to the rules of the JournalAntioxidants”. As requested, we activated the "Tracked Changes" option to highlight any changes in our paper.

The text was highlighted with yellow color to respond to some reviewers’ comments. Also, the text was highlighted in a turquoise color that mentioned the new additions.

As follows a clear list of our responses to Reviewers’ Comments, point by point.

Responses to Reviewer (2) Comments

  • The paper is well written and documented and the experimental part is meaningful.

Responses

Firstly, many thanks to the Reviewer (2) for describing our manuscript as “The paper is well written and documented and the experimental part is meaningful”.

  • Some aspects require rectification: table 1 is unreadable and it is useful to present some images with treated meat for evidentialist of the coating effect.

Responses

Thanks to Reviewer (2) for this comment.

As suggested, to clarify and explain Table 1, we added a new paragraph. Please see Page 5; Lines 196:234.

Several published articles used the same table to interpret the obtained results, such as:

Zalewska, M.; Górska-Horczyczak, E.; Marcinkowska-Lesiak, M. Effect of Applied Ozone Dose, Time of Ozonization, and Storage Time on Selected Physicochemical Characteristics of Mushrooms (Agaricus bisporus). Agriculture 2021, 11, 748. https://doi.org/10.3390/agriculture11080748

Marcinkowska-Lesiak, M.; PoÅ‚awska, E.; Wierzbicka, A. The effect of different gas permeability of packaging on physicochemical and microbiological parameters of pork loin storage under high O2 modified atmosphere packaging conditions. Food Science and Technology International 23(2) 174–184. https://doi.org/10.1177/1082013216671406

Regarding presenting some images with treated meat for evidentialist of the coating effect, we will give this suggestion full consideration for future studies.

Reviewer 3 Report

Introduction needs to be improved by adding more information about the current technical issues or problems that will be faced by the research (for instance, food waste) and a brief state of the art of other solutions applied recently (such as active packaging)

Add more thechnical details about the CMC used in the experimental part.

Try to express results included in Table 1 in a different way, for example including numbers and superscripts for ANOVA

Please discuss the optimal value of pH (at least a range) for consumers of the meat

Figure 2 and its methodology is unclear. Explain the absence of data (%wl) for the control sample. Perhaps it would be better to measure the weight loss during storage for each sample, taking the reference as the first day of measument (this is what one expects from the information included in section 2.6.2.)

Figure 3A 3B and 3C can be merged in a single Figure

Lipid oxidation and MetMb content are related to the same phenomenon, chemical changes due to oxygen. Perhaps both can be included in the same section and the results should better correlated

Please indicate if the current coated food sample would meet with regulations of food safety in terms of microbial growth and the maximum time of storage that one would expect at the tested conditions

Author Response

Responses to Reviewers’ Comments

First, many thanks to Editors and Reviewers for Recommending “Minor Revisions” to our manuscript for publication.

We provide upon resubmission:

Our revised manuscript was edited according to the rules of the JournalAntioxidants”. As requested, we activated the "Tracked Changes" option to highlight any changes in our paper.

The text was highlighted with yellow color to respond to some reviewers’ comments. Also, the text was highlighted in a turquoise color that mentioned the new additions.

As follows a clear list of our responses to Reviewers’ Comments, point by point.

Responses to Reviewer (3) Comments

  • Introduction needs to be improved by adding more information about the current technical issues or problems that will be faced by the research (for instance, food waste) and a brief state of the art of other solutions applied recently (such as active packaging).

Responses

As suggested, we improved supported the “Introduction Section”, and supported it with additional relevant citations [1,2,3,4,5,6,23,24,25]. Please see Page 2; Lines 49:57 and 78:83.

The relevant new references are properly cited within the text and in the list of references.

  • Add more technical details about the CMC used in the experimental part.

Responses

Thanks to Reviewer (3) for this comment.

As suggested, we added the requested details. Please see Page 3; Lines 122:127.

The relevant new reference is properly cited within the text and in the list of references.

  • Try to express results included in Table 1 in a different way, for example including numbers and superscripts for ANOVA.

Responses

Thanks to Reviewer (3) for this comment.

As suggested, to clarify and explain Table 1, we added a new paragraph. Please see Page 5; Lines 196:234.

Several published articles used the same table to interpret the obtained results, such as:

Zalewska, M.; Górska-Horczyczak, E.; Marcinkowska-Lesiak, M. Effect of Applied Ozone Dose, Time of Ozonization, and Storage Time on Selected Physicochemical Characteristics of Mushrooms (Agaricus bisporus). Agriculture 2021, 11, 748. https://doi.org/10.3390/agriculture11080748

Marcinkowska-Lesiak, M.; PoÅ‚awska, E.; Wierzbicka, A. The effect of different gas permeability of packaging on physicochemical and microbiological parameters of pork loin storage under high O2 modified atmosphere packaging conditions. Food Science and Technology International 23(2) 174–184. https://doi.org/10.1177/1082013216671406

  • Please discuss the optimal value of pH (at least a range) for consumers of the meat.

Responses

Thanks to Reviewer (3) for this comment.

As suggested, we added the requested information regarding the range of the optimal value of pH of the chicken meat. Please see Page 5; Line 238.

The relevant new reference is properly cited within the text and in the list of references.

  • Figure 3A, 3B and 3C can be merged in a single Figure.

Responses

Thanks to Reviewer (3) for this comment.

As suggested, we merged Figures 3 A, B, C, and D into a single Figure (Figure 3 “A to D”). Please see Page 7; Lines 303:315.

  • Lipid oxidation and MetMb content are related to the same phenomenon, chemical changes due to oxygen. Perhaps both can be included in the same section and the results should be better correlated.

Responses

Thanks to Reviewer (3) for this comment.

We prefer to keep lipid oxidation and MetMb content as two independent sections to be much more convincible “lipid in one section and protein in another section”.